# Substance and Behavioral Addictions among American Indian and Alaska Native Populations

**DOI:** 10.3390/ijerph19052974

**Published:** 2022-03-03

**Authors:** Claradina Soto, Amy E. West, Guadalupe G. Ramos, Jennifer B. Unger

**Affiliations:** 1Department of Population and Public Health Sciences, Keck School of Medicine, University of Southern California, Los Angeles, CA 90032, USA; ramosgua@usc.edu (G.G.R.); unger@usc.edu (J.B.U.); 2Department of Pediatrics, Keck School of Medicine, Children’s Hospital Los Angeles, University of Southern California, Los Angeles, CA 90027, USA; amwest@chla.usc.edu

**Keywords:** American Indians, Native Americans, Alaska Natives, substance use disorder, risk and protective factors, behavioral addictions, gambling addiction, gambling comorbidities

## Abstract

Objective: This paper examines substance and behavioral addictions among American Indian and Alaska Natives (AIAN) to identify the structural and psychosocial risk and cultural protective factors that are associated with substance use and behavioral addictions. Methods: Five databases were used to search for peer reviewed articles through December 2021 that examined substance and behavioral addictions among AIANs. Results: The literature search identified 69 articles. Numerous risk factors (i.e., life stressors, severe trauma, family history of alcohol use) and protective factors (i.e., ethnic identity, family support) influence multiple substance (i.e., commercial tobacco, alcohol, opioid, stimulants) and behavioral (e.g., gambling) addictions. Conclusions: There is a dearth of research on behavioral addictions among AIANs. Unique risk factors in AIAN communities such as historical trauma and socioeconomic challenges have interfered with traditional cultural resilience factors and have increased the risk of behavioral addictions. Future research on resilience factors and effective prevention and treatment interventions could help AIANs avoid behavioral addictions.

## 1. Introduction

Substance-related poisonings and deaths have increased among American Indians and Alaska Natives (AIAN) [1]. AIANs had the highest drug related deaths between 2013–2017, compared with other U.S. racial/ethnic groups [1]. Drug poisoning death rates for AIANs were 36.9 deaths per 100,000, while drug poisoning death rates for Whites were 32.6 per 100,000 during the aforementioned time period [1]. Commercial tobacco use is a significant correlate of morbidity and mortality and is more prevalent among AIAN than among other U.S. racial/ethnic groups [2].

Alcohol use is another salient problem in AIAN communities [3]. Alcohol-induced deaths were highest among AIANs, followed by Whites, and Latino groups, with 93.1, 12.9, and 12.2 per 100,000, respectively [4]. Deaths attributed to alcohol increased significantly at a rate of 4.0% per year from 2000–2017; this increase was also highest for AIANs compared to Whites and other ethnic and racial groups [1]. More recent analysis found that in 2019, the alcohol-involved death rate among AIANs was five times higher than that in the general population, at 50.5 deaths and 10.4 deaths per 1,000,000, respectively [3]. Moreover, this rate was 64% higher than it was in 2006, when the rate was 30.8 deaths per 100,000 [3].

Methamphetamine use in AIAN communities has also increased as a greater number of AIANs report that this is their drug of choice [5]. AIANs reported using methamphetamine at higher rates than heroin, marijuana, cocaine, and other drugs [6]. A more recent study found that approximately 15% of AIANs reported lifetime use of stimulants such as cocaine and methamphetamines [7].

Opioids are also a significant concern in AIAN communities. According to the 2018 National Survey on Drug Use and Health (NSDUH), 7.4% of AIAN adults ages 18–25 used opioids compared to 5.5% of adults in the general population [8]. The NSDUH also reported that opioid misuse rose from 4.7% in 2015 to 6.2% in 2018 for AIANs ages 26 and older [8]. Comparatively, 3.4% of adults ages 26 and older in the general population reported opioid misuse in 2018 [8]. These prevalence rates are likely underestimated by approximately 35% due to misclassification of AIAN race [9].

Although several articles have reported on the prevalence of substance use and other behavioral addictions in AIAN communities, the risk and protective factors within AIAN communities are not well understood. This article presents a narrative review on the state of the science on substance and behavioral addictions among AIANs. For this paper, substance use refers to alcohol, tobacco, opioids, and other drugs. The article describes the prevalence, risk factors, and protective factors for substance use addictions, followed by a brief presentation of behavioral addictions (e.g., texting, internet use, love, shopping, and gambling). We use a culturally centered approach, led by AIAN researchers, to synthesize the literature and identify the structural inequities and conditions that have led to a high prevalence of behavioral addictions as well as the cultural resilience factors that could potentially reduce them.

## 2. Method

### 2.1. Literature Review

The research team was guided by American Indian researchers (CS and AEW), who oversaw the process of gathering articles and synthesizing findings from a culturally centered perspective. The following search engines were utilized to compile the literature presented (1998–2021): PubMed, Science Direct, Cochrane Library, Ovid, and Google Scholar to identify studies related to substance use disorder and gambling addictions among American Indians and Alaska Natives (AIAN). Key words and search terms included the following combinations: (“American Indians” OR “Native Americans” OR “Alaska Natives”) AND (“substance use” OR “drug use” OR “substance use disorder” OR “behavioral addictions” OR “gambling addictions” OR “substance use and gambling comorbidity”) AND (“risk factors” OR “protective factors”). Additional studies were identified by checking reference lists of papers included in the review and by conducting manual searches of the authors found in the literature search. The search was limited to studies conducted in the United States because Indigenous populations in other countries have divergent histories of colonization and structural inequities.

A two-step process to review study inclusion was followed. First, article titles and abstracts were reviewed to identify whether (1) the articles had an AIAN sample exclusively or had a sample that included AIANs (2) articles examined substance use disorders (SUD) and/or behavioral or gambling addictions as outcomes. Other inclusion criteria were that articles had to be in English and used qualitative, quantitative, or mixed methods assessing SUD and/or behavioral addictions. Exclusion criteria included non-peer-reviewed sources. In the second step, authors reviewed the full-text articles previously identified and grouped studies according to the aforementioned outcomes. Authors resolved questions and disagreements through ongoing discussions and revisions to the matrix (described below). 

A total of 69 articles were utilized in the review and are described in Appendix A. Most of the literature consisted of cross-sectional designs along with systematic reviews, and qualitative studies. Appendix A includes studies that highlight the risk factors associated with substance use. Appendix A includes studies that highlight the protective factors associated with substance use and Appendix A includes behavioral addiction studies on gambling. Appendix A were developed to list all studies in the review and include study aim, study design, sample size (only applicable study), key findings, and measures used (only applicable study). If an article was a review, no samples size or measures were listed in Appendix A. The authors’ decisions to include case studies and several qualitative studies was influenced by wanting to honor Indigenous ways of knowing and Indigenous methodologies as this data highlights Indigenous voices and allows for community knowledge to be included [10].

### 2.2. Analysis 

This review was conducted using the matrix method as a guide to systematically organize and process literature according to content, methodologies, and outcomes [11]. In this method, literature is read at least twice, organized by subject/area, and once the matrix is finalized, the results are written according to this matrix. An iterative process is then followed to ensure that sources meaningfully contribute to the purpose of the review [11]. The updated integrative review guidelines [12] were also utilized to organize and incorporate sources from different methodologies (i.e., experimental, quasi-experimental studies) so that meaningful comparisons could be made. The combination of these methodologies was previously utilized in another literature review [13]. 

## 3. Results

### 3.1. AIAN Risk Factors for Substance Use Disorder

Risk factors are predictors or correlates of maladaptive behaviors or adverse conditions that exist at the structural, community, family, interpersonal, or individual levels [14] (Please see Appendix A). The presence of risk factors increases the chances that an individual will develop a disorder compared to someone else without this risk [15]. Protective factors are defined as beneficial variables that reduce the risk of adverse outcomes or interact with risk factors to buffer or mitigate the effects of risk [16]. It is important to identify both risk and protective factors because they represent opportunities for early intervention among those at greatest risk for adverse outcomes [14]. Risk and protective factors for substance use exist at multiple domains (e.g., individual, family, structural/socio-cultural) [14] and will be discussed accordingly. Historical trauma is a salient risk factor that pervades all levels of influence and will be discussed first.

### 3.2. Historical Trauma

Historical trauma describes the cumulative emotional and psychological wounding, over the lifespan and across generations, that occurs as the result of the massive group trauma experienced by AIAN peoples due to colonization and genocide incurred after European contact [17]. Historical trauma is a construct that has been richly debated and discussed in the literature; the narrative around historical trauma as a social and psychological construct is still evolving. For example, one recent article argued for the expansion of the construct to encompass the ongoing massive, structural violence perpetrated against AIAN people even in the current day [18]. Another recent article highlighted the need to engage in inquiry that helped capture parallel processes of individual versus collective meaning making related to historical trauma, to incorporate literature around cultural resilience and racial trauma, and to consider historical trauma within the lenses of both its metaphorical and practical/literal functions [19].

Historical trauma has long been conceptually linked to poor health outcomes in AIANs [20,21,22,23,24,25]. A recent systematic review (SR) examined the extent of empirical evidence that linked the experience of historical trauma to health outcomes and concluded that, while the literature is growing, it is still too disparate and inconsistent to make any coherent conclusions about the nature of the association between historical trauma and health outcomes, including substance use [19]. This SR revealed 19 studies of historical loss and health outcomes, 11 studies specifically examining residential school ancestry and health outcomes, and 3 “other” studies that did not fit neatly into either of the prior categories. Many studies reported significant associations between higher levels of reported historical trauma experience and adverse health outcomes; specifically, several studies linked historical trauma to increased substance use [26,27,28,29]. However, these studies suffered from various methodological weaknesses or differences that challenged the aggregation of their findings, thus limiting the extent to which they together can be used to form a cogent argument for the role of historical trauma in risk for the development of substance use disorders.

The authors noted that one of the most important areas for future research is to clarify and refine the theoretical and measurable construct of historical trauma in AIANs so that there is more consistency and comparability across studies. While coherence around a specific pattern or association remains unclear, the robust attention to historical trauma in the extant literature suggests that it is an important potential social determinant related to substance use and other health outcomes that should receive attention in future research. Gaining clarity around the role of historical trauma experience in risk or resilience for substance use behaviors has clear implications for the development and refinement of interventions to incorporate a trauma-informed approach and traditional healing practices as needed to address traumatic experiences and potential sequelae.

### 3.3. Sociocultural Risk Factors

Cultural norms can play a significant role in substance use among AIANs [30]. A cross-sectional survey found that AIAN adult males drink nearly three times as much alcohol as adult females [30]. Researchers assessed frequency and amount of alcohol consumed in open response format and findings indicated that males drank the equivalent of 1–2 glasses of wine once per week whereas females drank 1–2 glasses of wine once or twice per month [30]. One potential risk factor for this disparity is that historical and cultural norms have been more permissive of men drinking larger quantities of alcohol [30]. These norms are often passed on generationally, further perpetuating the pattern of alcohol abuse [30].

AIANs also experience acculturation related challenges, not only because of the transition from reservation to urban environments, but also because of the social demands to fit in with both the Native culture and American mainstream practices [31]. Whereas AIANs who live on reservations are immersed in an AIAN cultural context (at least while they are on reservation land), AIANs who live in urban areas must navigate a cultural context that differs from that of their ancestors [32]. The two sets of norms, values, and behaviors may be contradictory at times, and it may be challenging to integrate and adapt to both sets of norms and expectations [32]. Substance use can provide respite from the pressures associated with acculturation and balancing roles in minority and majority cultures [31]. Another sociocultural factor that plays a role in the onset and development of SUD is limited education about substance use and its adverse health effects in AIAN communities [33]. Limited information around SUD hampers individuals’ knowledge about cognitive behavioral strategies to prevent relapse and critically assessing drug use [34].

### 3.4. Poverty and Environment

Among AIANs, forced relocation, poverty-related stress, exposure to violence, and substance use are closely associated with each other and with substance use [30,33,35]. AIANs were forcibly removed from their lands to remote locations with stagnant rural economies that offered few opportunities to improve socio-economic status and living conditions [36]. The sparse job and economic opportunities, largely due to geographic remoteness and poor industrial diversity, perpetuated already high rates of poverty in various AIAN reservations, rancherias, villages and communities [36]. Despite substantial investment and increase in education among AIANs, the employment rate continues to decline, and wages have decreased [37]. Poverty-related financial stress intensifies family tensions, and some AIAN report that this is a major contributor to violence in the household [38,39]. Poverty also poses limitations on availability and accessibility of mental health or counseling services, potentially aggravating maladaptive coping strategies, including substance use [38]. Substance use is a common correlate of unemployment and poverty, further complicating individuals’ situational and relational dynamics [40,41]. While causal relationships cannot be drawn, it can be inferred that multiple poverty and violence related stressors contribute to and are closely tied to substance use [38].

In addition to forced relocation contributing to higher levels of poverty among many AIANs, it also promoted cultural displacement and eroded cultural practices and traditions [40,42]. [The cultural and emotional losses tied to forced relocation have been associated with increased SUD [40,42]. Urban environments away from reservations limited AIANs’ accessibility to social and family networks [39,43], increased alienation [44], exposure to discrimination, and potentially increased the risk of engaging in substance use [39].

### 3.5. Individual Risk Factors

#### 3.5.1. Biological

The use of alcoholic beverages is relatively new to AIAN cultures. Some tribes made beers and fermented drinks; however, they were typically consumed ceremonially and were not highly concentrated [45]. AIANs did not know about alcohol and were naïve about its effects prior to European colonists bringing it to Native lands and making the substance broadly accessible [45]. It has been theorized that AIANs are less likely to have the protective variant genes to metabolize alcohol. Reviewed studies indicate AIAN appear to lack metabolizing enzyme genes, similar to other population groups (East Asian and some Africans) (e.g., ALDH2*2 and ADH1B*3), but the effect of these genetic variants on SUD among AIAN has not been studied thoroughly [46]. AIANs started to consume alcohol after being introduced to it by colonists, but they had limited time to develop enzyme systems that facilitated alcohol metabolism [45]. It appears that biology is a potential risk factor for alcohol abuse in this population, as AIANs are less likely to have variations of genes that are essential to effectively metabolize alcohol [47]. However, it is unclear whether the higher prevalence of alcoholism among AIANs is related to differences in alcohol metabolism, or whether it is merely a response to trauma and limited opportunities. More research is needed to understand how AIANs metabolize other substances besides alcohol, as most of the available information is alcohol specific.

#### 3.5.2. Mental Health

Depressive symptoms and clinical depression are significantly associated with substance use among AIAN [30,48]. Psychosis, personality, and other mood disorders (e.g., posttraumatic stress disorder) are intricately linked with substance use and dependence as well [49,50,51]. Higher prevalence of mental health disorders is associated with more substance use, which contributes to an especially high public health burden of these dual diagnoses in AIANs. A study based on the National Epidemiologic Survey on Alcohol and Related Conditions found that 70% of AIAN men and 63% of AIAN females met the criteria for at least one mental health lifetime disorder as defined in the Diagnostic and Statistical Manual-IV [52]. In comparison, 62% of males and 53% of females in the general population met these criteria [52]. AIAN males and females had higher prevalence of both paranoid and antisocial personality disorders as well as drug dependence relative to both sexes in the general population [52]. Among AIANs, males had higher prevalence of schizoid personality disorder and AIAN females had higher prevalence of panic disorder [52]. The study also concluded that the prevalence for any substance use, mood, and personality disorders was higher for AIAN males and females compared to males and females in the general population [52]. It is important to consider, however, that there may be biases in the diagnostic process that may contribute to the elevated prevalence of mood and personality disorders among AIAN clients rather than actual higher prevalence of these disorders [53]. For instance, language and cultural variations in emotional and psychological expression, and Western definitions of mental disorders can affect diagnostic accuracy among AIANs, particularly when mental health providers are not familiar with the AIAN culture [53].

There are multiple potential reasons for the complex relationship between substance use and emotional and mental health disorders [52]. Some researchers explain that people experiencing depression self-medicate through substance use to cope and obtain relief from emotional pain [54]. Continually feeling stressed over challenging life events can exacerbate the impact of those stressors on the body and mind, outweigh coping resources and potentially make the use of substances a more appealing option [31,35,54,55]. Often indicative of great levels of despair and hopelessness, is the occurrence of suicidal thoughts and attempts, which are risk factors for SUD as well as potential results of substance use [44,56]. As can be inferred, risk factors often have synergistic and cumulative effects on SUD [50]. For instance, comorbid drug use, alcohol dependence, and psychiatric disorders exacerbate each other [57]. Cognitive-behavioral processes—specifically, positive substance-related expectancies pose a risk for SUD [33]. These positive expectancies—defined as individuals’ perceived outcomes about substances or their response to them—play a role, as individuals may underestimate their impact or believe that they will not be as severely affected by such substances [33]. Though definitive and causal explanations are challenging to assert, a substantial amount of research points to the role of individual coping abilities, self-efficacy, problem-solving skills, and cognitive-behavioral processes in the co-occurrence of substance use and mental illnesses.

#### 3.5.3. Polysubstance Use

Limited experimental and longitudinal research makes it difficult to assert whether polysubstance use is an outcome of substance use addictions or if polysubstance use is a risk factor for such addictions (or both), although there is some support for the latter [50,56,57]. What is known is that the use of alcohol typically precedes “harder drug” use (e.g., cocaine and methamphetamine). Additionally, early onset of substance use predicted subsequent and potentially higher rates of use [58,59], as individuals’ substance tolerance increases over time and early use sets up a pattern of addiction that may be challenging to stop. Strong relationships were found between past-year use of alcohol and drug dependence [50]. Similar findings were obtained in a study with AIAN veterans, where lifetime nicotine dependence was associated with lifetime alcohol use, drug use disorders, mood and anxiety disorders, and antisocial personality disorder [60].

Polysubstance use also appears to be tied to mental health. The comorbidities between nicotine dependence, psychiatric, and substance use disorders were studied with a nationally representative sample of AIAN adults [50]. Lifetime and past-year nicotine dependence were correlated with mood, anxiety, and personality disorders, alcohol use, and any drug use disorders [50]. Polysubstance use is also often linked to unresolved grief and ongoing discrimination [35].

#### 3.5.4. Physical Health

Physical health conditions play a substantial role in the onset and maintenance of SUD, particularly the presence of chronic medical conditions [56]. Medical illnesses and chronic pain can elicit stress and worry that can make individuals susceptible to SUD. Use of prescribed opioids for pain can lead to opioid addiction [61]. Experiencing acute physical ailments is also positively associated with frequently consuming large quantities of alcohol, an association that might be bidirectional [57]. Physical health conditions that adversely impact quality of life are risk factors for alcohol dependence [57]. Medical problems, particularly among AIANs living in reservations, are risk factors for SUD [56]. Accessibility to preventive healthcare and SUD treatment services is often limited [33,62], and IHS services not always being readily available outside of reservations [63] thwart timely and comprehensive care.

### 3.6. Family Risk Factors

Parental and other family members’ substance use was associated with increased likelihood of substance use among AIAN youth [48,64,65]. The role of social modeling, permissive attitudes toward substance use, diminished monitoring, and on rare occasions, parental offering of drugs and alcohol, provide possible explanations for the association between family substance use and SUD among AIAN youth [31,48]. Specifically, for some AIAN youth it is difficult to refuse drug or other substance offers when they come from parents and other family members or when tobacco products are offered during a ceremony [32,66]. Given the substantial emphasis that AIANs place on family relationships and the extended amount of time that AIAN youth spend with their family members, youth may be more hesitant to refuse engaging in substance use with their cousins or other adult relatives (e.g., Tribal elders), increasing substance use frequency [32]. The increased availability of substances in the home combined with fewer parental consequences for substance use can create opportunities for youth to explore different substances [32]. Moreover, wanting to belong in the family and cultivating strong social and family ties influence drug use as well [32].

Fathers’ alcohol consumption adversely impacts children as fathers can become neglectful [67]. Children with fathers who consume substances report that they do not want to be like their fathers or have similar adverse life trajectories and outcomes [67]. However, the limited availability of paternal role models who lead healthier life paths may leave youth without proactive coping strategies and render them more vulnerable to substance use [67]. Moreover, paternal substance use can lead to feelings of abandonment among children, potentially also contributing to substance use to alleviate the painful feelings [67].

Maternal mental illness and SUD is correlated with offspring’s substance use [59]. The quality of parenting skills is adversely impacted among parents who consume substances and/or who are suffering from anxiety and depression [68]. Detrimental parenting skills can contribute to further difficulties, as individuals may be subject to harsh or neglectful parenting practices when parents are using or are mentally ill [63]. Family discord and inharmonious parent-child relationships can also occur because of parental substance use and also contribute to their children’s SUD [64,65,69].

Strong associations have been observed between experiencing abuse in the family structure and substance use severity [51,58]. Dose-response relationships between adverse childhood experiences (ACEs) and multiple risk factors have been documented among AIANs and non-AIANs [51,70]. Per the Warne and colleagues’ study, AIANs who had six or more ACEs (e.g., neglect,) were significantly more likely to experience mood and anxiety disorders along with alcohol and nicotine use [51]. Another study on the association between ACEs and substance use among AIANs found that sexual abuse nearly doubled the odds of alcohol and other drug use [58]. The same study also found that witnessing family violence significantly contributed to alcohol use [58]. While causal relationships cannot be asserted in these domains, research indicates that traumatic events in the family were independently associated with various forms of substance use [58]. Parental divorce, family separation, and parental loss was also associated with alcohol use and other substances [58].

Historical context should also be considered when discussing family-related risk factors. Family bonds and parental attachments were severed when AIANs were sent to boarding schools and these occurrences hindered AIANs’ ability to trust, communicate and build relationships with loved ones [69]. Attachment injuries were passed on generationally, complicating the process of successfully navigating interpersonal relationship challenges and potentially led to substance use [69]. Forced relocation and boarding school separations left many generations of AIANs without their parents, and consequently, without good parenting role models or the support of multigenerational families [69]. Hence, family structures, parenting abilities, and supportive environments need to be rebuilt for the future generations’ bonds and well-being.

### 3.7. AIAN Protective Factors for Substance Use Disorders

#### 3.7.1. Sociocultural and Community Protective Factors

Connecting AIANs with their culture, language, traditions, and heritage through elders is protective because these interactions help to develop cultural knowledge, strength, and increase cultural identity [71] (Please see Appendix A). In turn, cultural identification and embracing cultural characteristics, norms, and traditional values can increase community cohesiveness, foster wellness, and reduce distress among AIAN [35,54]. Healing practices build spiritual identity and promote health behaviors [72]. For instance, Wellbriety is a peer-led, effective intervention for SUD and other emotional disorders that incorporates various protective factors including connecting to and enhancing individuals’ spirituality, building on ancestors’ resilience, and involving emotional, mental, physical, and spiritual dimensions in all aspects of health [73]. The focus on culturally sensitive beliefs and practices have made Wellbriety a particularly fitting and widely-used intervention that holds promise for addressing AIANs’ SUD needs [73]. Additionally, cultural practices such as storytelling to enhance resilience are also protective, as they often remind AIANs of their ancestors, their resilience, and the value of their practices [74,75].

#### 3.7.2. Individual Protective Factors

Characteristics such as the ability to find a greater meaning, purpose, and internal strengths build and contribute to resilience among AIAN [71]. Additionally, focusing on positive aspects of people’s own lives, and their community and culture promote well-being among Native populations [76]. Emotion regulation, valuing ethnic identity, empowerment, and activism also promote mental health in this population [76]. Decision making and problem-solving skills also appear to be protective for substance use disorder (SUD) [48] as individuals may be able to best evaluate the risks and benefits of using substances and then find more adaptive activities. Self-efficacy is also protective against SUD possibly by trusting that they can work through an obstacle or improve their mood, for instance.

#### 3.7.3. Family Protective Factors

Social support and supportive family and peer relationships with people who help strengthen cultural identity are protective for SUD [54]. Additionally, continuous parental involvement and positive interactions help AIAN youth make healthier decisions around substance use [54]. Drinking with family members or peers who drink lower amounts of alcohol compared to socializing with people who drink a lot can also be protective against developing a more severe SUD [48]. Family cohesion, parental attachment, and low family conflict are also protective [44,77] perhaps because of the ability to discuss hardships and process upsetting emotions with others and thereby avoid resorting to substance use. Developmentally appropriate family rules and restrictions also protect youth from SUD by potentially providing structure and monitoring as indicated in the general parenting literature [78].

### 3.8. Behavioral Addictions

#### 3.8.1. Background

Behavioral addictions are defined as dysregulated behaviors aimed to satisfy appetitive needs that can result in unwanted consequences [79]. Behavioral addictions reflect difficulties with impulse control as individuals repeatedly engage in maladaptive behaviors that are often accompanied by significant impairment [80,81]. Appetitive-related behaviors can occur on a continuum; repetitive engagement in such behaviors despite negative consequences qualifies them as addictions [79]. The satisfactory, yet deceptive feelings about the appetitive need being fulfilled are short-lived and a new cycle of need, behavior, illusory fulfillment, and adverse outcome ensues [79]. Literature reviews on the prevalence of addictions in the US adult population suggest that nearly 50% of Americans had past-year indicators of addictive behaviors [81,82].

Little is known about behavioral addictions among AIAN populations. A recent study on AIAN adolescents in California found that the most prominent lifetime addictions were texting, internet use, love, and shopping, while cigarette use, texting, internet use, and love (defined as maladaptive relationships that mirror addictive behaviors) were the most prevalent past-month addictions [81,83]. The number of substance and/or behavioral addictions increased with age in this sample of AIAN youth [83]. Social acceptance, depression, and the number of negative life events were significant predictors of substance and/or behavioral addictions [83]. These findings underscore the need for additional research on the comorbidity of substance and behavioral addictions as well as the occurrence of multiple behavioral addictions.

#### 3.8.2. Gambling Addiction

Gambling appears to be the most thoroughly studied addiction among AIANs (Please see Appendix A); however, this literature lacks specific data on the protective factors associated with this addiction. As such, our discussion of gambling addiction will primarily focus on its prevalence and associated risk factors. The limited studies on gambling that do exist have primarily been conducted in only a few geographic areas. The available research on behavioral addictions indicates that AIANs are at increased risk for problem and pathological gambling, relative to other racial and ethnic groups in the United States [84]. A study on two North Dakota Indian reservations found the pathological gambling rate was 14% and 10%, respectively, compared to 6% statewide [84]. A second study of AI in New Mexico found the prevalence of pathological gambling was significantly higher for AI (2.2%) than among non-AIANs (0.9%) [85]. A third study showed significantly higher rates among AIAN veterans (9.9%) compared to Hispanic veterans (4.3%) from southwest and north central regions of the USA [86]. These three localized studies highlight the varied pathological and problem gambling rates among AIAN adults, and unfortunately are all consistently higher than their counterparts. One national study on AIAN in the U.S. indicated AIANs also had higher problem gambling rates compared to the general population (18% vs. 8%, respectively) [87].

The increased risk for gambling addiction among AIAN might be due to other inequities faced by AIAN that are also risk factors for problematic gambling. These inequities and risk factors include low education, unemployment, few socioeconomic opportunities, untreated historical trauma, higher rates of substance use, and having a casino nearby that serves as the main employer and source of revenue for the community [88,89,90,91]. Because many AIAN Tribes operate casinos on their sovereign land to earn income for the Tribe, AIANs could be more likely to live near casinos, work at casinos, or view casinos as a beneficial part of their community. A casino could be lucrative to a Tribe overall but harmful to some individual members if they are at increased risk for problematic gambling, especially among a population that has few other economic opportunities. A study of risk factors for gambling among a large sample of AIAN and Blacks across the U.S. found that neighborhood disadvantage and readily available opportunities for gambling were significantly associated with overall gambling, frequent gambling, and problem gambling [87]. Researchers also found that lower socioeconomic status was significantly associated with problem gambling [92].

Other studies have pointed to the associations among abuse, mental health problems, and gambling addictions [93,94]. A study with survivors of child sexual abuse in residential schooling found that survivors had unresolved grief, and that was linked to pathological gambling [93]. Another study found a strong association between problem gambling and past-year psychiatric disorders among AIAN compared to non-AIANs [94].

## 4. Discussion

This article provides a review of addictions among AIANs, but the sparse literature on this population only includes substance use and gambling addiction. Other behavioral addictions probably exist in this community, as in any community, but almost no research exists. More research attention is needed to learn about other potential behavioral addictions. Substance use and gambling addictions are significant concerns among the AIAN population. Consistent with prior research in non-AIAN samples, poverty, acculturation, and personality disorders [95,96,97,98] appear to be risk factors for addiction in AIAN individuals as well [33,43,50]. Multiple risk factors contribute to the higher prevalence of these addictions, with many of these factors arguably tracing back to the historical trauma experienced by AIANs. Previous research has established historical trauma as directly linked to poor health outcomes and increased substance use [20,21,22,26,27,28,29,99]. The repercussions of historical trauma continue to be evidenced in current structural racism and social determinants of health that adversely impact AIANs’ overall health and outweigh their coping resources [27,33,43]. Mental and physical health challenges are largely related to restricted availability of culturally sensitive care and accessibility of healthcare services. Family related risk factors can also be traced back to severed family relationships that occurred with forced separation and boarding school experiences. Family bonds, healthy parenting role models, consistent socialization, the provision of extended family’s love and affection, and cultural teachings were virtually eliminated with AIANs’ removal from their lands and social networks [17,54,100,101]. With forced removals, AIANs lost their social support and valuable resources, and had to start anew with very limited assets and means of sustenance [22,35]. These experiences left many AIANs with the pressures of finding new ways of living in new remote areas that had limited opportunities for growth and with multiple reasons for wanting to potentially escape what they were experiencing and depleting their emotional well-being and coping abilities. Potentially as a direct result, or through indirect associations, AIANs resorted to substance use and other behavioral addictions to cope with the multiple, chronic, and acute stressors related to discrimination, cultural and material losses, genocide, and knowledge of the deliberate efforts to erase the culture [26,27,28,29,99].

Despite the numerous injustices and inequities that AIANs experienced, and continue to experience, AIANs fostered and held on to cultural values and practices that help build resilience. Developing a strong ethnic identity, and holding and passing on knowledge of the culture, traditions, and languages through generations has been particularly protective among this population. Engaging in culturally healing practices (e.g., sweats) and utilizing medicines (e.g., sage, sweet grass) support recovery and promote well-being [24,102,103]. Reconnecting with the Native culture, building internal strengths, coping resources, and relying on extended social networks and multi-generational relationships assist AIANs with navigating the historical and every day hardships experienced [102,104,105].

There are promising culturally appropriate addiction treatment programs that incorporate AIAN healing practices and traditions in SUD services [106]. One example of a substance use treatment intervention specifically for AIANS is the Drum-Assisted Recovery Therapy for Native Americans (DARTNA) [106,107,108]. This drumming intervention has been shown to benefit AIANs in recovery, enhance cognition, and decrease physical ailments [106,107,108]. Another program that utilized culturally tailored telepsychiatry in a residential treatment setting was also effective in promoting treatment engagement and improving completion rates among Alaska Natives [109]. Attendees of such programs have shown higher levels of spiritual connectedness and significant reductions in depression and anxiety [106]. These programs have also provided clients opportunities to reconnect with their culture and their community [110]. These indigenous approaches are based on indigenous centered priorities that include AIANs to develop frameworks based on community values and perspective. Programs developed by the AIAN people for the AIAN community allow for methodologies that fit AIAN ideologies rather than being confined to Westernized frameworks.

## 5. Future Directions

There is much work to be done to ameliorate the mental, physical, and social damages that the AIAN population experiences. The path to move forward must include interventions that capitalize on AIANs’ strengths and resilience. These interventions and initiatives must be done in a culturally respectful and sensitive manner. Future research must also recognize the ecological and historical context for addictive behaviors among AIANs. Future work must also be based on appropriate treatment options that incorporate cultural traditions and ways of healing. Research with the AIAN population must also be done with humility and willingness to follow cultural practices, such as listening to and engaging the AIAN community and not imposing Westernized frameworks and methodologies when they are not well-received. AIAN communities know what recovery and appropriate treatment looks like, and as such, their lead should be followed. Future studies should also examine other behavioral addictions, and ideally should utilize large, diverse, representative samples for their findings to be generalizable. Understanding the gambling addiction and why AIANs engage in gambling can support future interventions to reduce gambling urges. There is also great need for longitudinal studies to confidently establish risk and protective factors, as many of the available studies are cross-sectional where the direction of causality is not clear. Future studies should examine the role of education among AIANs and the associations to the onset and development of SUD. There is a need to address and understand the traumatic experiences between AIAN older and younger generations and its impact to the rates of SUD problems. Lastly, there are probably other risk and protective factors that have yet to be identified to understand SUD problems in AIAN populations and deserve more attention.

## Data Availability

Not applicable.

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
