# Peer review of "Substance and Behavioral Addictions among American Indian and Alaska Native Populations"

_ijerph, 2022, doi:10.3390/ijerph19052974_

Round 1

Reviewer 1 Report

The authors have answered and addressed the comments I had in the previous round.

Reviewer 2 Report

Dear Authors,

I would like to congratulate you with the manuscipt you've submitted.

The update version is really good.

Kind regards

TU

Reviewer 3 Report

Dear authors,

The manuscript has been significantly improved.

This manuscript is a resubmission of an earlier submission. The following is a list of the peer review reports and author responses from that submission.

Round 1

Reviewer 1 Report

This is a review article about SUB in AIAN. As in many countries the native people have higher levels of alcohol dependence other substance use disorders. At the end the authors discuss the lack of research for this subpopulation in North America with a very unique history. I have a few comments that I hope will contribute to improving the manuscript

The description of the search method should be improved and clarified. I suspect that all key words were not included simultaneously, as “substance use” was repeated.

To provide a better overview of the result of the search I would suggest listing all selected publication in a Table. In that table important characteristics of the publications can also be presented. I leave it up to the authors to decide, but I was thinking of sample size, study population (proportion AIAN), and which outcomes and associations that where assessed in each study.

Discussion

I understand that the authors would like to focus on AIAN as this group has a very special history. However, when it comes to the paths to SUD or protective factors these might very well be shared with the population in general. Is the unique history the main difference between AIAN and the general population? Or are there also difference in protective factors, for example greater meaning and purpose, that are mentioned in 3.7.2

If one what to avoid imposing a Westernized framework or methodologies, is it possible to compare the situation for AIAN with native people in other countries? Is there any research made that could be discussed or inspired of for further research?

Minor:

The abbreviation SUD appears before it is written out.

Reviewer 2 Report

Dear Authors,

thank you for having chance to review you manuscript.

I have several concerns regarding presenting topics that, from my opinion, don't allow the manuscript to be published.

Authors present the interesting topic of increased risk for substance abuse in AIAN group. They claim substance-related poisonings and deaths have increased among American Indians and Alaska Natives (AIAN). Moreover, alcohol-induced deaths were highest among AIANs, followed by Whites, and Latino groups. Methamphetamine use in AIAN communities has also increased as a greater number of AIANs report that this is their drug of choice. Additionally, opioids are also a significant concern in AIAN communities.

Authors claim the importance of identification both risk and protective factors because they represent opportunities for early intervention among those at greatest risk for adverse outcomes. The authors noted that one of the most important areas for future research is to clarify and refine the theoretical and measurable construct of historical trauma in AIAN so that there is more consistency and comparability across studies.

Despite all the presented information, I have some important issues that raise my concern that the thesis presented in the manuscript may be incorrect:

1. Authors say that AIANs experienced acculturation related challenges, not only because of the transition from reservation to urban environments, but also because of the social demands to fit in with both the Native culture and American mainstream practices.

My concern: Are you sure that this is the main reason? Can you present the studies comparing the early experience of AIANs who were transferred and presented higher substances addition comparing to those, living relatively long in new society and presented lower substances addiction? Is this the real cause of they higher substances- related poisoning?

 2. Authors present thesis that sociocultural factor that plays a role in the onset and development of SUD is limited education about substance use and its adverse health effects in AIAN communities.

My concern: can you present the differences between high and low educated individuals in AIAN group? Is your theory presented based on subjective opinion or deep reaserch?

3.Among AIANs, forced relocation, poverty-related stress, exposure to violence, and substance use are closely associated with each other and with substance use. AIANs were forcibly removed from their lands to remote locations with stagnant rural economies that offered few opportunities to improve socio-economic status and living conditions.

My concern: Is the next generation of AIANs presenting different pattern? Can we link their behavor to presented thesis or maybe to minority predisisposition?

4. AIANs started to consume alcohol after being intro- duced to it by colonists, but they had a limited time to develop enzyme systems that facil- itated alcohol metabolism. It appears that biology is a potential risk factor for alcohol abuse in this population as AIANs are less likely to have variations of genes that are es- sential to effectively metabolize alcohol.

My concern: Authors present AIANs as humans who can not properly judge what is good and what is not. From my humble point of view, its humiliating for this minority and offensive. They are not children and each men can judge and realize what is right and what is wrong. I have strange suggestion as authors treat AIANs as humans of second category. Those evaluation raise my concern humans rights.

5. AIAN males and females had higher prevalence of both paranoid and antisocial personality disorders as well as drug dependence relative to both sexes in the general population [48]

Can you distinguish if those prevalences have changed after AIANS became part of modern society?

6. Social support and supportive family and peer relationships with people who help strengthen cultural identity are protective for SUD [51]. Additionally, continuous parental involvement and positive interactions help AIAN youth make healthier decisions around substance use [51]. Drinking with family members or peers who drink lower amounts of alcohol compared to socializing with people who drink a lot can also be protective against developing a more severe SUD [44]. Family cohesion, parental attachment, and low family conflict are also protective [41, 76, 77] perhaps because of the ability to discuss hardships and process upsetting emotions with others and thereby avoid resorting to substance use.

Authors present supportive effect of family/relationship support. Still, I’m not convinced that presented initial habits were related to AIANs stress related to modern society demands or to their historical background.

7. Gambling appears to be the best studied addiction; however, this slightly larger literature lacks specific data on the protective factors associated with this addiction. As such, our discussion of gambling addiction will primarily focus on its prevalence and associated risk factors. The limited studies on gambling that do exist have primarily been conducted in only a few geographic areas. The available research on behavioral addictions indicates that AIANs are at increased risk for problem and pathological gambling, relative to other racial and ethnic groups in the United States.

My concern: Gambiling was related to AIANs desire to become rich in easy way? Explaining the addiction by simple AIANs historical background is questionable.

Kind regards

Reviewer 3 Report

  1. About the abstract:

- Purpose of the study: in simple words should tell the readers about the objective of this study. There is no discussion, no history, just the objective of this study and this aspect is not clearly identified.

- Methodology: it should clearly appear methods, software, review and survey that has been used to do this study.

- Main findings do not appear: Write only the main findings in a few words.

- It is convenient to highlight the applications of this study: where this study can be useful, give name of area, disciplines, etc.

- I recommend describing the Novelty/Origin of this study: what is new in this study that can benefit the readers and how it is advancing existing knowledge or creating new knowledge on this topic.

  1. Introduction and Literature Review. The background of the study is presented clearly, but with an inadequate amount of background to the study; it is recommended that it be included as current and relevant references that give greater strength to the theoretical framework.

  1. Methodology. This section does not meet the minimum requirements. It does not include a section on the "Material and methods" used. Without this section it is very difficult to understand how the comparative analysis was carried out and, fundamentally, to accept its scientific rigor. In this sense, it is necessary to define the research design used, the criteria of validity and qualitative reliability, and the specific procedure followed in the content analysis that, it is understood, was intended to be carried out. Without this section, the analysis falls within the scope of opinion based on non-systematic readings of normative texts. Mention all research conditions, assumptions, theories followed. It does not include methodological orientation, sampling [number of sources, method of approach, improvement of data collection] and data analysis.

  1. Discussion. A logical and scientific analysis of the results of the study is not provided. No evidence is presented to support your analysis by citing the work of previous researchers or existing theories. The combination of its findings in relation to those previously identified in the literature review, and placing them within the context of the theoretical framework that supports the study, does not appear. On the other hand, the limitations of the study are reported, as well as the proposed measures derived from the research. The conclusions section includes some information in this regard, but in a very superficial manner.